# Trends in Complementary Feeding Indicators and Intake from Specific Food Groups among Children Aged 6–23 Months in Bangladesh

**DOI:** 10.3390/ijerph19010550

**Published:** 2022-01-04

**Authors:** Sabuj Kanti Mistry, Md Belal Hossain, Nafis Md Irfan, Manika Saha, Silvia Saberin, Abu Ahmed Shamim, Amit Arora

**Affiliations:** 1BRAC James P Grant School of Public Health, BRAC University, 68 Shahid Tajuddin Ahmed Sharani, Mohakhali, Dhaka 1212, Bangladesh; belal.hossain@ubc.ca (M.B.H.); aashamim@gmail.com (A.A.S.); 2Centre for Primary Health Care and Equity, University of New South Wales, Sydney, NSW 2052, Australia; 3ARCED Foundation, 13/1 Pallabi, Mirpur-12, Dhaka 1216, Bangladesh; 4Department of Public Health, Daffodil International University, Dhaka 1207, Bangladesh; 5School of Population and Public Health, University of British Columbia, 2206 East Mall, Vancouver, BC V6T1Z3, Canada; 6Institute of Nutrition and Food Science, University of Dhaka, Dhaka 1000, Bangladesh; nafis.irfan35@gmail.com; 7Interdisciplinary Graduate Program in Human Toxicology, University of Iowa, Iowa City, IA 52246, USA; 8Department of Internal Medicine, University of Iowa, Iowa City, IA 52246, USA; 9Department of Human-Centred Computing, Faculty of Information Technology, Monash University, Clayton, VIC 3800, Australia; manika.saha@monash.edu; 10Independent Researcher, Dhaka 1212, Bangladesh; sseve25@gmail.com; 11Translational Health Research Institute, Western Sydney University, Campbelltown Campus, Penrith, NSW 2751, Australia; a.arora@westernsydney.edu.au; 12School of Health Sciences, Western Sydney University, Campbelltown Campus, Penrith, NSW 2751, Australia; 13Oral Health Services, Sydney Local Health District and Sydney Dental Hospital, NSW Health, Surry Hills, NSW 2010, Australia; 14Discipline of Child and Adolescent Health, Sydney Medical School, Faculty of Medicine and Health, The University of Sydney, Westmead, NSW 2145, Australia; 15Heath Equity Laboratory, Campbelltown, NSW 2560, Australia

**Keywords:** complementary feeding, infant and young child feeding, minimum acceptable diet, minimum dietary diversity, minimum meal frequency

## Abstract

The present study aims to comprehensively analyse trends in complementary feeding indicators (Introduction of solid, semi-solid, and soft foods at 6–8 months (INTRO), Minimum Dietary Diversity (MDD), Minimum Meal Frequency (MMF) and Minimum Acceptable Diet (MAD)) among children aged 6–23 months in Bangladesh. The study used data from four rounds (2007, 2011, 2014, and 2017–2018) of nationally representative Bangladesh Demographic and Health Surveys (BDHSs). The Cochran–Armitage test was performed to capture the trends in complementary feeding practices and intake from specific food groups. BDHSs are periodically conducted cross-sectional surveys in all seven administrative divisions of Bangladesh. The present analysis was performed among 8116 children (1563 in 2007, 2137 in 2011, 2249 in 2014, and 2167 in 2017–2018) aged 6–23 months. Overall, a decreasing trend was observed in all the complementary feeding indicators except INTRO from 2007 to 2014, but a substantial increase in MDD, MMF and MAD was noted in 2017–2018. A statistically significant reduction in consumption from different food groups such as legumes and nuts (*p* < 0.001), dairy products (*p* = 0.001), vitamin-A-rich fruits or vegetables (*p* < 0.001), and other fruits and vegetables (*p* < 0.001) was also observed. However, a positive trend was noted in the consumption of grains/roots/tubers (*p* = 0.027), and meat/fish/egg (*p* < 0.001). After experiencing a significant decreasing trend during 2007–2014, the recent BDHS indicates improvements in all complementary feeding indicators among young children in Bangladesh, which calls for integrated, multisectoral, and multicomponent interventions to sustain this progress.

## 1. Introduction

Malnutrition is among the significant public health challenges among young children in low- and middle-income countries. According to the World Health Organization, around 45% of the deaths are attributed to nutrition-related factors among children aged less than five years [1]. The first 1000 days of life is recognised as the ‘critical window’ for optimal growth and development for a child [2]. It is well established that initiation of breastfeeding within the first hour of birth and exclusive breastfeeding to 6 months of life can protect the infant from various infectious diseases and reduce the risk of neonatal mortality [3]. However, children are at an increased risk of malnutrition after 6 months of life as, at this stage, the mother’s milk alone is no longer sufficient to meet the energy and nutritional requirements for infants [4]. This transition from exclusive breastfeeding to family foods, starting around the period of 6 months of age and continuing until 24 months, is referred to as complementary feeding [3]. There is an estimate that 6% of all deaths in children under the age of 5 years in developing countries can be prevented if appropriate complementary feeding practices are followed [5]. Therefore, timely introducing complementary feeding and giving nutritionally adequate and safe complementary foods starting at the period of six months is recommended by the World Health Organization (WHO) [6].

Late introduction and inadequate quantity of complementary feeding can result in faltered physical growth and cognitive development among infants [7,8]. Additionally, inappropriate complementary feeding practices have been found as a determinant of increased malnutrition outcomes (such as wasting, stunting and underweight) and under-5 child mortality worldwide [9,10]. If a child remains malnourished during this period, it results in a compromised immune system and a higher risk of severe infectious diseases including diarrhoea and pneumonia [11,12]. Malnutrition during these young ages can also lead to irreversible adverse effects for the rest of their lives and they may never meet their full physical and intellectual potential [8,13,14]. 

In Bangladesh, malnutrition continues to be a serious public health problem among young children and still, around one-third of children aged under five years are stunted [15]. Late introduction and inappropriate complementary feeding practices are among the major causes of child malnutrition in Bangladesh [16]. Many underlying and immediate causes of malnutrition are related to food insecurity and inappropriate feeding practices, which are often compounded by socio-economic and political contexts [8]. Previous studies have shown different socio-economic and individual factors influence complementary feeding practices among young children in Bangladesh [16,17,18]. In addition, factors such as women’s empowerment, exposure to mass media, and antenatal care practices have been associated with young children’s dietary diversity [19]. 

Considering the importance of infant and young child feeding (IYCF) practices, the Government of Bangladesh has incorporated IYCF in many national health policies and programs such as the health population and nutrition sector development program HPNSDP (2011–2016) [20], second National Plan of Action for Nutrition (NPAN2) (2016–2025) [21] and National Strategy for IYCF in Bangladesh (2007) [22]. In addition, The National Women Development Policy 2011 and National Labour Policy 2012 provide extended maternity leave to support IYCF to their children [23]. Further, implementation of the Baby-Friendly Hospital Initiative, enforcing the Breast Milk Substitutes (BMS) (Regulation of Marketing) Act, helps to protect young children from commercial baby foods and to promote improved IYCF, including exclusive breastfeeding [24].

Bangladesh demographic and health surveys (BDHSs) are periodically conducted nationwide data, which follow a well-established methodology and incorporate large numbers of covariates of IYCF. A recent study [25] analysed four rounds of BDHS data from 2004 to 2014 and found stagnating trends in complementary feeding practices in young children. As new BDHS (2017–2018) data are available, a national-level analysis with updated information will provide essential and updated information for the stakeholders. Therefore, the present study has been undertaken to analyse the trends in complementary feeding practices among children aged 6–23 months using data from the most recent four rounds of BDHSs, including the most recent one (2007, 2011, 2014 and 2017–2018). The study presents the trends in different complementary feeding indicators and explains the trends in intake of complementary foods from different food groups. Findings from this study will contribute to implementing important and effective initiatives by programme and policymakers to improve complementary feeding practices to combat widespread malnutrition among young children in Bangladesh. 

## 2. Materials and Methods

### 2.1. Data Sources

Data from the four most recent rounds (2007, 2011, 2014, and 2017–2018) of BDHSs were used in this study. These nationally representative cross-sectional surveys followed a two-stage stratified random sampling design and covered all seven administrative divisions of Bangladesh. The detailed methodology of the respective surveys can be found in the BDHS reports [15,26,27]. BDHS 2007, 2011, 2014, and 2017–2018 collected information from 10,996, 17,749, 17,863, and 20,127 ever-married women aged 15–49 years, respectively, with a response rate of around 98%. Among these 66,735 selected women from four rounds of surveys, 61,716 were usual residents (de jure population), and the remaining were non-residents to certain geographical areas. BDHSs usually collect information about mothers and their children born within five years preceding the survey. For the present analysis, we considered information of a total of 8116 children (1563 in 2007, 2137 in 2011, 2249 in 2014, and 2167 in 2017–2018) aged 6–23 months (see Figure 1).

### 2.2. Outcome Measures

We measured the trends in complementary feeding practices using the four indicators recommended by the World Health Organization (WHO) [28]. These include Introduction of solid, semi-solid and soft foods (INTRO); Minimum Dietary Diversity (MDD); Minimum Meal Frequency (MMF); Minimum Acceptable Diet (MAD) for infant and young children. Mothers of children were interviewed regarding these indicators and were asked to recall foods given to their children in the last 24 h preceding the survey date. Brief definitions of the four outcome variables (as per WHO recommendations) are given below; these outcome variables were measured as per the given definitions:Timely introduction of solid, semi-solid and soft foods (INTRO): the proportion of children between 6 and 8 months of age who received solid, semi-solid or soft foods.Minimum dietary diversity (MDD): the proportion of children who received at least four or more food groups out of six food groups. Instead of seven food groups recommended by the WHO guidelines, the complementary food items provided in the last 24 h were classified into six food groups in the BDHS data as flesh foods, and eggs were combined as one group. These six groups include (1) grains, roots, and tubers; (2) legumes and nuts; (3) dairy products; (4) meat/fish/egg; (5) vitamin-A-rich fruits and vegetables; and (6) other fruits and vegetables. It is notable to mention that egg intake was captured separately from the 2011 BDHS. Therefore, trend analysis for egg intake during 2011 to 2017–2018 are also presented in this study.Minimum meal frequency (MMF): the proportion of children aged 6–23 months (breastfed and non-breastfed children) who received solid, semi-solid or soft foods the minimum number of times or more in the previous day of the survey. Minimum meal frequency was defined as having at least 2 meals/day for 6–8 months, 3 or more meals/day for 9–23 months old, breastfed children and 4 or more meals/day for non-breastfed children.Minimum acceptable diet (MAD): the proportion of children aged 6–23 months who received and satisfied both the conditions of minimum dietary diversity and minimum meal frequency.

### 2.3. Explanatory Variables

Socio-demographic variables considered in this study were: administrative division (Barisal, Chittagong, Dhaka, Khulna, Rajshahi, Rangpur, Sylhet, and Mymensingh); place of residence (urban/rural); household wealth quintile (poorest, poorer, middle, richer, and richest); child age (6–8 months, 9–11 months, 12–17 months, 18–23 months); child gender (male/female); birth order (1, 2, ≥3); maternal age at child’s birth (<20 years, 20–29 years, ≥30 years); maternal education (no education, grade 1–4, 5–9, and 10 or more); maternal occupation (housewife/working outside); fathers’ education (no education or grade 1–4, 5–9, and 10 or more); fathers’ occupation (services holder/business; agriculture-based work, non-agriculture-based work, and others). 

The household wealth index was constructed using factor analysis by the DHS team and categorized as poorest, poorer, middle, richer, and richest. The details can be found in BDHS reports [15,26,27]. Body mass index (BMI) for mothers was calculated as weight in kg/(height in meter)^2^, which was categorized using the Asian cut-off for underweight (BMI < 18.5), normal (18.5–22.9), overweight (23.0–24.9), and obese (≥25.0) [29].

### 2.4. Statistical Analysis

To assess the distribution of the different variables, we performed descriptive analysis. The Cochran–Armitage test [30,31] was performed to determine the trends of complementary feeding practices among different categories of covariates. All tests were performed at a 5% level of significance. Sampling weights were used in all analyses to ensure the actual representation of the nationwide data. All analyses were performed using the statistical software package STATA (Version 13.0).

## 3. Results

### 3.1. Trends in Complementary Feeding Practice

The trends in INTRO among 6–8 months aged children and MDD among 6–23 months aged children in Bangladesh from 2007 to 2018 are shown in Table 1 and Figure 2. The proportion of children aged 6–8 months who received solid, semi-solid or soft foods decreased from 72.8% to 68.0% during the period of 2007 to 2014 and increased to 75.0% in 2017–2018. However, this increasing trend was not statistically significant (*p* = 0.505). Additionally, a nearly steady but insignificant trend in INTRO among children from all groups was observed.

The trends of MDD among 6–23 months aged children are also shown in Table 1 and Figure 2. The proportion of MDD reduced from 41.4% to 27.3% from 2007 to 2014 and then increased to 56.7% in 2017–2018, resulting in a significantly increasing trend (*p* < 0.001). Additionally, a significantly increasing trend was observed in all strata except that of the Rajshahi Division, where a not significantly increasing trend was noted.

The MMF decreased from 79.9% to 61.8% from 2007 to 2014 and after that increased to 78.7% in 2017–2018 (Table 2 and Figure 2). Overall, it showed a decreasing trend, but was not statistically significant (*p* = 0.935). A substantial reduction in MMF was observed among children from the Dhaka and Rajshahi divisions, those aged 18–23 months, those whose mother had no or primary education or were underweight, and those whose father was an agriculture-based worker, unemployed/student, retired, or a beggar. It was also noted that MMF was significantly increased in Chittagong and Rangpur Division, those whose mother was overweight, and those whose father had completed secondary education and was involved in non-agriculture-based work.

The MAD decreased from 39.4% in 2007 to 23.1% in 2014 and then significantly increased to 52.1% in 2017–2018, resulting in a significantly increasing trend (*p* < 0.001) (Table 2 and Figure 2). Additionally, a significantly increasing trend was observed in all strata except that of the Rajshahi Division and those whose father was involved in non-agriculture-based work. 

### 3.2. Trends in Consumption from Different Food Groups

Overall trends in the consumption of food items from six food groups are shown in Figure 3. The consumption of grains, roots, or tubers significantly decreased from 84.6% to 78.2% from 2007 to 2014 but increased to 87.4% in 2017–2018. Although a significant increase in meat/fish/egg consumption (46.9% to 67.2%) was observed, there was a substantial reduction in consumption of legumes or nuts (29.7% to 21.5%), dairy products (39.7% to 32.2%), vitamin-A-rich fruits or vegetables (53.8% to 37.7%), and other fruits or vegetables (46.4% to 26.6%) from 2007 to 2017–2018. 

The consumption of grains/roots/tubers significantly reduced among children from the Rajshahi division but increased among children from the Chittagong and Sylhet divisions, those living in urban areas, those from poorer and the richest households, and those whose mother only had primary or secondary education (Table 3). On the other hand, the consumption of legumes/nuts decreased over time for all children regardless of age, gender, place of residence, wealth status, and parents’ education and occupation. 

A significant reduction in consumption of dairy products was observed among children regardless of age, gender, place of residence, wealth status, and parents’ education and occupation (Table 4). On the other hand, meat/fish/egg consumption increased over time among all children regardless of age, gender, place of residence, wealth status, and parent’s education and occupation. 

Table 5 shows the trends in consuming vitamin-A rich fruits or vegetables and other fruits or vegetables among children aged 6–23 months in Bangladesh from 2007 to 2018. The consumption of vitamin-A-rich fruits/vegetables and other fruits/vegetables decreased over time among all children regardless of age, gender, place of residence, wealth status, and parent’s education and occupation. 

## 4. Discussion

The present analyses were performed by pooling data from the four most recent rounds of nationally representative Demographic and Health Surveys of Bangladesh. Along with identifying trends in complementary feeding indicators, this study also investigated trends in dietary consumption from different food groups. Overall, a significantly increasing trend was observed for MDD and MAD, which may partially be attributable to the recent enactment of several IYCF policies in Bangladesh, such as the BMS Act 2013 [24] and NPAN 2 [21]. Consumption of food items from different food groups (grains, roots, tubers; legumes or nuts; dairy products; vitamin-A rich fruits or vegetables; other fruits or vegetables) decreased over time apart from an increasing trend in grains/roots/tubers and meat/fish/egg consumption. Divisional disaggregated data revealed the most remarkable reducing or non-significant trend in the Rajshahi division. These findings draw further attention to policymakers and program implementers to ascertain the probable reasons and inputs such as strengthening health systems and channelling behaviour change communication to be provided in this region to reduce regional variability and vulnerability.

While comparing complementary feeding practices across different strata, the most significant reduction was observed among children from the poorest wealth quintiles. Rural children were more vulnerable to a greater reduction in MDD and MAD, while a more substantial reduction in INTRO was observed among urban children in Bangladesh. One possible explanation could be that urban people have more purchasing power than rural families, and with the increase in income, children’s diets become more diversified [19,32]. However, there is also counterevidence (analysis from BDHS 2007 and 2014) revealing that children’s dietary diversity is achieved more in rural regions of Bangladesh [19]. The plausible reasons could be the easy access that rural people have to varied foods through different activities such as homegrown food production, livestock raising and fish-pond culture [33]. 

While exploring complementary feeding practices across child characteristics, no substantial gender differences were observed. However, the time trend analyses showed a more significant reduction in INTRO and MDD among boys than girls. It is also important to note that the achievement of an increasing trend in MDD, MMF, and MAD was lowest among children aged 6–8 months compared to older children (12–17 months and 18–23 months). Therefore, there is a need to focus on younger children, particularly when complementary feeding is initiated, and greater attention is required when designing feeding programs and policies. We found that children from the higher birth order achieved an increasing trend in INTRO, MDD, MMF, and MAD compared to the first birth order, implying that as mothers have more children, it becomes easier for them to achieve optimal feeding practices. Similar findings are also reported in other studies conducted in Ethiopia and India, where increased birth order is associated with the better achievement of INTRO, MDD and MAD [34,35]. These findings suggest that mothers become more experienced by nurturing their first child and perform better complementary feeding practices for subsequent children. 

Although a significant improvement in feeding indicators in all education groups in 2017–2018 was observed, the current study findings revealed that higher maternal education was associated with the better achievement of INTRO, MDD, MMF, and MAD during 2007–2014. A similar trend was also observed for paternal education. These findings are corroborated by many other national studies across different regions such as Sri Lanka [36], India [37], Tanzania [38], and Nigeria [9]. A possible explanation of this relationship is that education helps mothers improve their knowledge on appropriate infant and young child feeding practices [39,40]. Thus, the universal coverage of primary education and the Bangladesh government’s initiative to provide increased access to higher education for girls resulted in better compliance with complementary feeding guidelines. 

Maternal employment has an equivocal impact on the complementary feeding practice of young children due to women empowerment. In contrast, there is well-documented evidence suggesting that maternal employment leads to time constraints for mothers, who allocate less time for cooking diverse foods [41,42]. The current study revealed a greater reduction in INTRO and less improvement in MMF and MDD among working mothers compared to housewives. Therefore, it is pertinent to support working mothers with initiatives such as setting up breast-friendly workplaces or arranging childcare centres near their workplace. Our study findings also identified children of younger mothers (aged <20 years) who reported lower achievement in INTRO, MDD and MAD compared to older mothers (>30 years) as younger mothers may need additional support in achieving optimal feeding practices. Similar findings have been reported in the literature in nationally representative studies from Nepal and India [35,43]. Therefore, policymakers need to prioritize young mothers in delivering appropriate complementary feeding messages.

A decreasing trend was observed between 2007 and 2018 in the consumption of foods from different food groups such as legumes and nuts; dairy products; vitamin-A-rich fruits or vegetables; and other fruits or vegetables while consumption of grains/roots/tubers and meat/fish/egg was significantly increased during this time. These trends in consumption were consistent regardless of the age, gender, place of residence, wealth status of the children as well as the occupation and education of their parents. In contrast, trend analysis (2001–2014) from Nepal reported a reduction in the consumption of grains, roots, and tubers while an increased consumption in other five food groups among young children [32]. If compared with other South Asian countries, children from Bangladesh had higher consumption of meat and dairy products compared to those from Afghanistan, Nepal, and Pakistan, while consumption of vitamin-A-rich fruits and vegetables is similar to that of Afghanistan, Sri Lanka and Maldives [1]. It is notable to mention that there is a possibility of seasonal variation in fruits/vegetable consumption as BDHSs are carried out at different times of the year. It is also worthy to note that some BDHSs covered both summer and winter seasons, and the respondents were asked to report the consumption of both fruits and/or vegetables; thus, the seasonal variation may be minimized.

Overall, the study attempted to underpin different indicators of complementary feeding practices and consumption of various food groups in children aged 6–23 months in Bangladesh across different times (from 2007 to 2018) and how these indicators changed across different strata with time. Findings of the present study impart several suggestions such as a strong focus on young children, intensive training to young mothers on optimal feeding practices, supporting working mothers with setting up breastfeeding-friendly workplaces or childcare facilities at or near workplaces, and considering the divisional, regional (rural/urban), birth order, and parental education variability in complementary feeding practices while designing and implementing feeding programs and policies. Despite improvements in women’s education, increased per capita income, reduced child marriages, increased access to health care services, and antenatal visits, trends in complementary feeding practices in children aged 6–23 months have been declining in Bangladesh until 2014. An upward trend has been noted in 2017–2018 BDHS in some of the indicators and appropriate attention is needed to sustain the progress. More effort is required to ascertain how adherence to complementary feeding guidelines can be improved and future well-designed studies are warranted to elucidate the causality and factors associated with complementary feeding practices.

While the study’s main strength is the use of data from nationally representative surveys that employed well-established methods and standardised questionnaires to collect information, the study was also subjected to several limitations. Firstly, the BDHSs questionnaire measures complementary feeding indicators using 24 h recall data, which may lead to recall bias and social desirability bias by mothers, providing an under- or over-estimation of indicators. Secondly, in measuring association and predicting causality, a further well-designed longitudinal study considering potential confounding variables is required. However, the present study has generated valuable findings using large-scale nationally representative survey data that still have program implications. Moreover, it is also notable to mention that a previous publication presented the trends in feeding practices using the same survey for the period of 2004–2014. Yet, we have added the latest round of survey in our analysis (2017–2018). 

## 5. Conclusions

The study reported a decreasing trend in almost all complementary feeding indicators and consumption from different food groups among children aged 6–23 months in Bangladesh from 2007 to 2014. However, the recent BDHS (2017–2018) noted improvements in all complementary feeding indicators. Policymakers and practitioners need to focus on an integrated, multicomponent, and multisectoral intervention to sustain this progress, and focus should be given to behaviour change communication targeting the most vulnerable populations.

## Figures and Tables

**Figure 1 ijerph-19-00550-f001:**
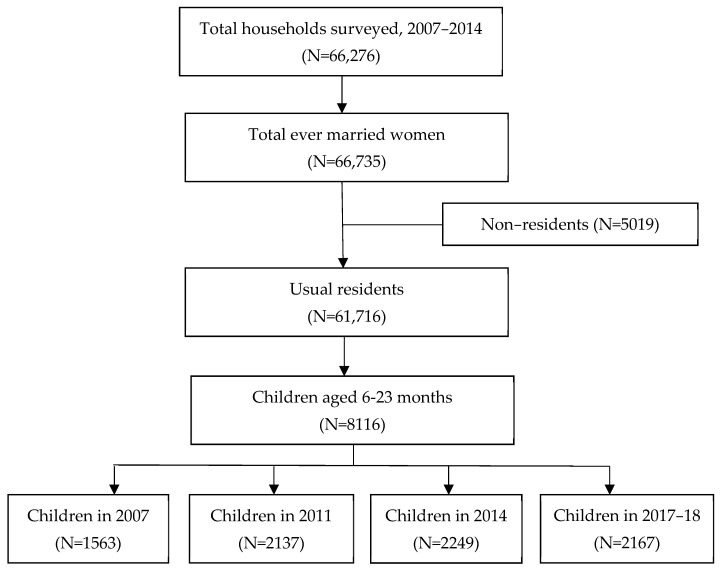
Study profile and participants enrolment.

**Figure 2 ijerph-19-00550-f002:**
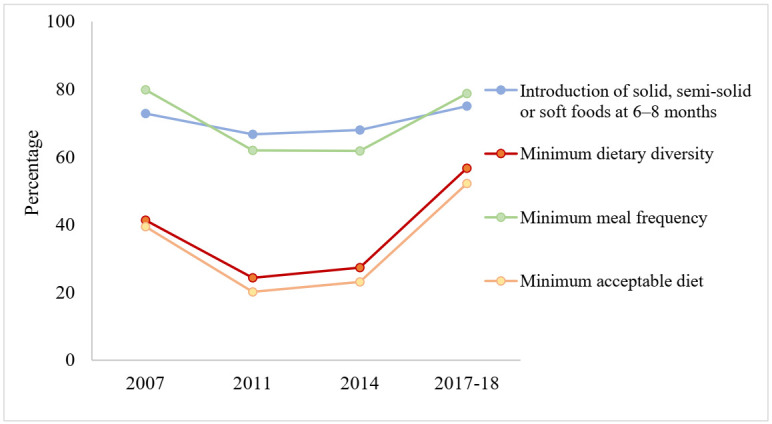
Trends of recommended infant and young child feeding (IYCF) practices among 6–23 months aged children in Bangladesh, 2007–2018.

**Figure 3 ijerph-19-00550-f003:**
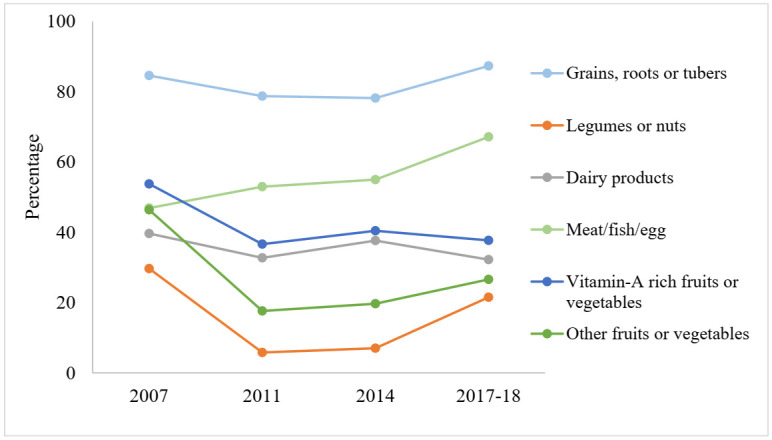
Trends of intake from different food groups among 6–23 months aged children in Bangladesh, 2007–2018.

**Table 1 ijerph-19-00550-t001:** Trends of introduction of solid, semi-solid or soft foods (among 6–8 months aged children) and minimum dietary diversity (among 6–23 months aged children) in Bangladesh, 2007–2018.

Characteristics	Introduction of Solid, Semi-Solid or Soft Foods (%)						Minimum Dietary Diversity (%)					
	2007	2011	2014	2017–2018	% Change ^3^	*p* ^4^	2007	2011	2014	2017–2018	% Change ^3^	*p* ^4^
Overall	72.8	66.7	68.0	75.0	3.0	0.505	41.4	24.3	27.3	56.7	37.1	<0.001
Household characteristics												
Administrative division												
Barisal	69.9	60.5	77.8	66.0	−5.6	0.868	35.9	23.1	29.6	54.7	52.6	0.002
Chittagong	52.1	55.4	56.6	69.8	33.9	0.056	35.4	20.8	22.4	57.2	61.7	<0.001
Dhaka	79.8	68.4	76.3	87.2	9.4	0.156	42.3	25.4	28.4	54.3	28.4	<0.001
Khulna	79.1	90.1	76.6	80.1	1.3	0.673	45.7	31.1	36.2	59.1	29.3	0.003
Rajshahi	81.9	77.7	81.6	69.0	−15.8	0.180	51.9	30.9	27.3	57.0	9.8	0.834
Rangpur ^1^	-	74.8	59.7	69.3	−7.3	0.493	-	23.4	28.7	66.2	183.3	<0.001
Sylhet	66.4	50.6	47.3	59.4	−10.5	0.467	24.7	14.2	24.2	52.8	114.2	<0.001
Mymensingh ^2^	-	-	-	77.2	-	-	-	-	-	52.5	-	-
Place of residence												
Urban	81.8	76.0	68.6	80.5	−1.6	0.809	45.4	34.9	32.6	60.1	32.4	<0.001
Rural	70.6	64.6	67.7	72.8	3.0	0.544	40.2	21.1	25.5	55.4	38.1	<0.001
Household wealth status												
Poorest	71.0	52.8	60.1	63.6	−10.5	0.581	32.2	12.4	18.0	48.4	50.4	<0.001
Poorer	60.9	64.3	75.3	69.4	13.9	0.129	32.0	16.8	22.3	54.8	71.6	<0.001
Middle	77.1	71.5	63.0	80.8	4.7	0.988	45.2	25.5	28.4	56.4	24.7	0.001
Richer	80.5	70.1	77.4	70.8	−12.1	0.408	45.4	33.2	31.8	54.6	20.4	0.006
Richest	83.6	81.9	65.6	91.2	9.1	0.783	54.2	37.0	37.3	70.1	29.5	<0.001
Child characteristics												
Age (month)												
6–8	72.8	66.7	68.0	75.0	3.0	0.505	9.0	6.4	5.8	19.8	118.7	<0.001
9–11	-	-	-	-	-	-	31.8	19.6	19.6	43.8	37.9	0.015
12–17	-	-	-	-	-	-	49.1	27.8	27.9	66.5	35.4	<0.001
18–23	-	-	-	-	-	-	56.1	33.7	42.1	70.2	25.0	<0.001
Gender												
Male	76.7	66.2	69.5	76.7	0.0	0.774	42.4	24.6	26.0	57.5	35.7	<0.001
Female	69.5	67.3	66.3	72.8	4.8	0.658	40.4	24.1	28.8	55.8	38.2	<0.001
Birth order												
1	82.3	74.4	66.9	76.4	−7.2	0.205	48.3	30.1	33.1	62.2	28.6	<0.001
2	73.5	67.1	71.3	78.1	6.2	0.290	41.6	25.1	24.8	54.0	29.7	<0.001
≥3	65.1	60.8	65.8	68.9	5.9	0.467	35.1	18.7	22.8	53.5	52.4	<0.001
Maternal characteristics												
Age at child’s birth (years)												
<20	73.6	63.6	62.7	71.5	−2.9	0.659	40.4	27.5	28.0	55.2	36.8	<0.001
20–29	74.0	69.1	69.6	73.1	−1.2	0.909	43.0	23.4	27.2	57.5	33.7	<0.001
≥30	68.5	61.7	69.7	86.1	25.7	0.027	37.3	24.0	26.9	55.8	49.8	<0.001
Level of education												
0–4	63.0	54.9	62.5	63.1	0.2	0.822	31.5	15.7	15.8	47.1	49.7	<0.001
5–9	77.6	72.1	71.6	75.8	−2.4	0.760	47.5	25.5	28.1	56.7	19.5	<0.001
10+	95.2	83.0	71.1	85.3	−10.4	0.303	55.9	43.9	45.1	67.4	20.6	<0.001
Occupation												
Housewife	70.2	65.5	68.8	76.0	8.2	0.163	40.8	23.8	27.4	55.6	36.5	<0.001
Working outside	81.2	87.3	64.7	73.4	−9.5	0.127	43.4	31.7	27.2	58.3	34.5	<0.001
Body mass index (kg/m^2^)												
<18.5	78.9	56.8	63.2	73.5	−6.8	0.207	37.6	20.0	22.0	57.5	53.1	<0.001
18.5–22.9	70.2	71.2	67.0	69.9	−0.4	0.688	40.2	24.6	27.1	54.4	35.3	<0.001
23.0–24.0	66.5	58.4	71.5	81.9	23.2	0.047	48.2	28.9	32.5	60.6	25.9	<0.001
≥25.0	69.8	80.7	77.3	82.0	17.4	0.483	63.8	34.0	34.5	58.4	−8.5	0.053
Paternal characteristics												
Level of education												
0–4	69.2	61.0	62.2	69.0	−0.2	0.798	34.1	17.7	20.0	50.5	47.9	<0.001
5–9	78.4	67.1	69.8	75.2	−4.0	0.715	43.4	25.5	28.4	55.2	27.2	<0.001
10+	72.3	85.2	77.5	83.1	14.9	0.410	60.3	38.9	40.1	69.1	14.6	<0.001
Occupation												
Service holder/businessman	73.8	77.9	70.5	72.2	−2.2	0.513	47.8	29.3	33.5	60.6	26.6	<0.001
Agriculture-based worker	78.6	62.7	73.6	68.5	−12.8	0.327	40.1	20.6	22.2	52.4	30.7	<0.001
Non-agriculture-based worker	73.6	63.3	70.1	80.8	9.8	0.037	46.2	26.1	28.8	56.0	21.4	0.009
Others ^5^	63.5	59.4	48.2	68.4	7.8	0.709	29.5	15.1	20.1	56.1	90.0	<0.001

^1^ The administrative division Rangpur was created in 2010; ^2^ The administrative division Mymensingh was created in 2015; ^3^ “+” represents increase, “-” represents decrease between 2007 and 2018; ^4^
*p*-value for the Cochran–Armitage test of % change between 2007 and 2018; ^5^ Unemployed/student, retired, beggar, etc.

**Table 2 ijerph-19-00550-t002:** Trends of minimum meal frequency and minimum acceptable diet among 6–23 months aged children in Bangladesh, 2007–2018.

Characteristics	Minimum Meal Frequency (%)						Minimum Acceptable Diet (%)					
	2007	2011	2014	2017–2018	% Change ^3^	*p* ^4^	2007	2011	2014	2017–2018	% Change ^3^	*p* ^5^
Overall	79.9	62.0	61.8	78.7	1.4	0.935	39.4	20.2	23.1	52.1	32.2	<0.001
Household characteristics												
Administrative division												
Barisal	72.3	61.2	60.0	74.7	−3.3	0.688	32.7	17.3	24.4	49.2	50.2	0.004
Chittagong	65.7	52.2	54.6	75.4	−14.8	0.005	32.5	16.2	17.3	49.9	53.8	<0.001
Dhaka	83.0	62.9	60.3	79.1	4.7	0.025	39.7	22.2	25.0	51.3	29.2	<0.001
Khulna	88.4	80.2	79.2	82.6	6.6	0.211	44.7	28.0	31.6	54.5	21.9	0.022
Rajshahi	91.3	62.5	70.8	73.4	19.6	<0.001	51.6	24.8	25.8	52.0	0.7	0.235
Rangpur ^1^	-	70.6	65.4	85.6	−21.3	<0.001	-	19.2	23.5	63.6	230.7	<0.001
Sylhet	72.3	55.8	57.4	76.2	−5.4	0.407	23.8	11.3	17.9	47.1	98.1	<0.001
Mymensingh ^2^	-	-	-	84.7	-	-	-	-	-	49.1	-	-
Place of residence												
Urban	83.2	65.4	65.9	80.7	3.0	0.931	43.5	28.4	27.4	55.4	27.3	<0.001
Rural	78.9	60.9	60.3	78.0	1.1	0.840	38.2	17.7	21.6	51.0	33.3	<0.001
Household wealth status												
Poorest	83.6	52.9	53.7	75.6	9.6	0.098	31.8	10.2	14.7	44.6	40.3	<0.001
Poorer	76.2	61.8	61.1	79.4	−4.3	0.293	31.7	13.8	17.5	50.5	59.3	<0.001
Middle	78.8	66.8	63.0	77.1	2.2	0.518	43.2	22.4	23.2	50.0	15.7	0.030
Richer	82.0	61.5	65.9	78.6	4.2	0.935	42.9	27.3	28.1	50.9	18.4	0.006
Richest	78.9	69.4	66.4	83.1	−5.3	0.305	49.3	30.1	33.1	65.5	32.9	<0.001
Child characteristics												
Age (month)												
6–8	59.9	51.3	48.8	62.2	−3.8	0.804	9.0	6.3	5.8	19.3	115.0	<0.001
9–11	73.8	57.2	52.3	70.3	4.7	0.286	28.7	14.9	16.1	38.9	35.8	0.002
12–17	83.8	62.7	67.2	84.1	−0.4	0.063	46.8	23.3	24.4	62.0	32.5	<0.001
18–23	89.7	70.2	68.0	85.0	5.2	0.036	54.1	27.9	34.6	63.8	17.9	<0.001
Gender												
Male	80.6	62.9	62.9	78.1	3.1	0.634	39.9	20.4	22.7	51.8	29.9	<0.001
Female	79.1	61.0	60.6	79.4	−0.4	0.722	39.0	20.0	23.6	52.5	34.6	<0.001
Birth order												
1	82.5	64.6	64.2	82.6	0.0	0.764	45.4	26.0	27.8	57.7	27.2	<0.001
2	80.0	62.8	61.7	75.9	5.1	0.621	40.0	19.0	20.6	49.1	22.7	<0.001
≥3	77.5	59.0	58.9	77.5	−0.1	0.839	33.9	16.0	19.9	49.1	45.1	<0.001
Maternal characteristics												
Age at child’s birth (years)												
<20	82.4	62.7	61.5	77.9	5.4	0.072	39.3	23.4	22.9	50.8	29.3	0.008
20–29	79.4	62.2	62.5	78.3	1.3	0.917	40.6	19.5	23.1	52.5	29.3	<0.001
≥30	78.0	60.5	60.0	80.4	−3.1	0.116	35.8	18.9	23.5	52.3	46.2	<0.001
Level of education												
0–4	77.5	54.6	55.2	75.1	3.1	0.010	30.6	13.3	12.7	42.5	38.9	<0.001
5–9	82.0	63.7	63.1	77.6	5.4	0.574	44.6	20.8	23.7	52.0	16.5	0.015
10+	81.0	75.5	69.4	85.6	−5.6	0.107	53.0	37.2	39.7	63.3	19.3	<0.001
Occupation												
Housewife	78.9	61.4	60.8	76.3	3.3	0.076	38.7	19.7	22.6	49.7	28.5	<0.001
Working outside	83.2	68.8	65.1	82.5	0.8	0.754	42.1	26.7	24.8	55.9	33.0	<0.001
Body mass index (kg/m^2^)												
<18.5	83.8	61.2	61.6	78.8	6.0	0.001	36.6	17.0	18.1	51.5	40.8	<0.001
18.5–22.9	78.1	62.3	59.6	76.9	1.6	0.356	38.3	20.2	22.8	49.9	30.0	0.008
23.0–24.0	73.4	60.1	65.0	82.5	−12.4	<0.001	44.8	22.8	27.1	58.2	30.1	<0.001
≥25.0	79.1	65.1	67.3	80.2	−1.3	0.048	58.0	28.7	30.7	53.3	−8.0	0.036
Paternal characteristics												
Level of education												
0–4	79.8	56.7	55.3	78.5	1.6	0.074	33.6	15.1	16.1	47.1	40.1	<0.001
5–9	80.1	64.8	64.2	75.9	5.2	0.453	41.4	21.5	24.3	49.2	18.9	<0.001
10+	79.7	69.7	70.0	84.1	−5.5	0.035	54.1	30.7	35.0	65.2	20.7	<0.001
Occupation												
Service holder/businessman	78.9	66.4	63.1	79.3	−0.6	0.848	44.4	24.5	27.9	56.2	26.6	<0.001
Agriculture-based worker	83.1	62.3	57.2	78.3	5.8	0.005	39.9	17.9	18.8	47.7	19.5	0.200
Non-agriculture-based worker	78.2	60.8	66.9	79.5	−1.7	0.002	43.0	21.1	25.5	51.9	20.6	<0.001
Others ^5^	79.2	54.3	54.6	75.0	5.4	0.007	28.8	11.9	15.3	50.3	74.3	<0.001

^1^ The administrative division Rangpur was created in 2010; ^2^ The administrative division Mymensingh was created in 2015; ^3^ “+” represents increase, “-”represents decrease between 2007 and 2018, respectively; ^4^
*p*-value for the Cochran–Armitage test of % change between 2007 to 2018, respectively; ^5^ Unemployed/student, retired, beggar, etc.

**Table 3 ijerph-19-00550-t003:** Trends of consuming grains, roots or tubers, and legumes or nuts among 6–23 months aged children in Bangladesh, 2007–2018.

Characteristics	Grains, Roots or Tubers (%)						Legumes or Nuts (%)					
	2007	2011	2014	2017–2018	% Change ^3^	*p* ^4^	2007	2011	2014	2017–2018	% Change ^3^	*p* ^5^
Overall	84.6	78.8	78.2	87.4	3.2	0.027	29.7	5.8	7.0	21.5	27.5	<0.001
Household characteristics												
Administrative division												
Barisal	82.1	77.8	79.9	84.2	2.6	0.524	34.9	9.3	9.9	22.9	34.3	0.045
Chittagong	76.0	73.4	75.4	88.6	16.7	<0.001	26.7	3.6	6.3	24.6	7.9	0.568
Dhaka	86.1	76.4	75.0	84.7	−1.6	0.289	33.7	5.9	9.5	21.5	36.1	<0.001
Khulna	90.7	89.4	86.0	89.6	−1.3	0.613	23.8	10.2	6.6	17.1	28.2	0.116
Rajshahi	92.5	83.8	85.4	86.1	−7.0	0.012	30.5	5.7	3.2	15.2	50.2	<0.001
Rangpur ^1^	-	87.5	85.4	90.2	3.0	0.384	-	6.1	2.6	17.3	−181.4	<0.001
Sylhet	77.0	72.7	74.7	88.8	15.3	0.010	22.1	4.8	5.8	30.8	−39.7	0.031
Mymensingh ^2^	-	-	-	89.5	-	-	-	-	-	24.4	-	-
Place of residence												
Urban	85.6	80.8	80.3	90.4	5.6	0.032	39.1	7.6	10.0	24.6	37.1	<0.001
Rural	84.4	78.2	77.5	86.2	2.2	0.212	26.9	5.3	6.0	20.4	24.1	<0.001
Household wealth status												
Poorest	86.3	76.0	79.6	88.1	2.1	0.188	24.6	3.5	4.3	19.6	20.6	0.144
Poorer	80.2	78.9	75.5	88.5	10.3	0.011	25.2	6.1	5.4	18.5	26.7	0.014
Middle	86.2	80.0	76.5	83.9	−2.6	0.325	28.4	5.5	5.5	18.5	34.7	0.001
Richer	85.2	78.7	80.1	84.3	−1.1	0.986	29.3	6.8	9.3	21.5	26.8	0.094
Richest	85.8	81.1	79.0	92.0	7.3	0.045	41.9	7.6	10.8	30.1	28.1	0.003
Child characteristics												
Age (month)												
6–8	56.0	47.9	39.7	56.0	0.0	0.465	9.1	4.2	2.6	8.3	9.4	0.475
9–11	86.6	79.4	76.7	89.8	3.7	0.397	24.2	4.4	7.1	19.9	17.8	0.434
12–17	91.4	84.2	87.2	93.3	2.0	0.035	32.8	6.7	5.7	23.6	27.9	0.006
18–23	93.0	90.5	88.8	94.7	1.8	0.458	40.3	6.6	10.7	26.4	34.6	<0.001
Gender												
Male	85.2	78.5	78.5	87.4	2.5	0.129	28.5	5.4	6.8	21.0	26.1	0.002
Female	84.0	79.1	77.9	87.3	3.9	0.112	30.9	6.3	7.3	22.1	28.5	<0.001
Birth order												
1	85.1	79.6	81.0	87.2	2.4	0.228	34.0	6.7	7.4	23.3	31.3	<0.001
2	85.2	81.0	77.8	88.0	3.3	0.318	30.7	6.4	6.8	20.4	33.6	0.003
≥3	83.8	76.3	75.0	86.9	3.7	0.764	25.3	4.5	6.8	20.7	18.1	0.031
Maternal characteristics												
Age at child’s birth (years)												
<20	84.0	77.7	79.9	84.6	0.7	0.764	29.3	4.9	7.7	19.7	32.9	<0.001
20–29	84.5	79.1	78.5	87.4	3.5	0.063	29.9	6.3	6.7	23.1	22.7	0.001
≥30	86.0	79.0	75.3	89.4	3.9	0.174	29.4	5.1	7.4	18.8	36.1	0.015
Level of education												
0–4	82.7	74.4	71.6	85.1	2.9	0.670	27.1	3.7	5.8	18.5	31.6	<0.001
5–9	86.5	80.0	80.4	87.6	1.3	0.171	29.6	6.8	7.6	21.8	26.5	0.037
10+	85.4	86.1	82.7	89.3	4.5	0.213	38.4	7.9	7.4	24.2	36.9	0.009
Occupation												
Housewife	83.2	78.5	77.5	86.2	3.5	0.190	31.1	5.6	7.4	23.1	25.8	<0.001
Working outside	89.4	82.2	80.4	89.2	−0.2	0.796	25.0	7.9	5.9	19.1	23.4	0.070
Body mass index (kg/m^2^)												
<18.5	87.1	78.4	76.4	86.9	−0.2	0.175	28.7	5.9	7.0	23.7	17.4	<0.001
18.5–22.9	83.7	79.3	78.2	86.5	3.4	0.188	28.9	5.4	6.2	20.4	29.6	<0.001
23.0–24.0	76.8	76.5	78.9	88.1	14.7	0.001	31.9	6.9	7.8	20.5	35.9	0.622
≥25.0	87.5	79.9	81.0	89.2	1.9	0.071	38.7	6.6	9.3	23.1	40.3	0.608
Paternal characteristics												
Level of education												
0–4	85.1	77.3	76.2	86.3	1.4	0.944	26.7	5.4	6.7	18.3	31.6	<0.001
5–9	83.9	78.1	78.2	87.1	3.8	0.047	30.7	4.9	6.9	22.3	27.4	0.057
10+	84.6	83.8	82.3	89.5	5.9	0.051	37.0	8.5	8.0	25.2	32.0	0.029
Occupation												
Service holder/businessman	84.2	81.3	77.8	85.6	1.7	0.808	29.6	6.2	7.4	21.9	26.2	0.023
Agriculture-based worker	85.1	79.8	79.5	86.5	1.7	0.907	28.2	5.1	3.7	20.9	25.9	<0.001
Non-agriculture-based worker	86.9	77.4	79.9	88.8	2.2	0.009	36.3	6.8	8.9	22.5	38.1	0.005
Others ^5^	81.8	74.9	72.9	87.1	6.5	0.652	23.2	3.1	7.4	18.2	21.4	0.005

^1^ The administrative division Rangpur was created in 2010; ^2^ The administrative division Mymensingh was created in 2015; ^3^ “+” represents increase, “-“represents decrease between 2007 and 2018; ^4^
*p*-value for the Cochran–Armitage test of % change between 2007 and 2018; ^5^ Unemployed/student, retired, beggar, etc.

**Table 4 ijerph-19-00550-t004:** Trends of consuming dairy products and meat/fish/egg among 6–23 months aged children in Bangladesh, 2007–2018.

Characteristics	Dairy Products (%)						Meat/Fish/EGG (%)					
	2007	2011	2014	2017–2018	% Change ^3^	*p* ^4^	2007	2011	2014	2017–2018	% Change ^3^	*p* ^4^
Overall	39.7	32.7	37.6	32.2	18.7	0.001	46.9	53.0	55.0	67.2	43.2	<0.001
Household characteristics												
Administrative division												
Barisal	40.4	25.5	35.7	25.1	38.0	0.090	43.0	51.4	57.6	65.9	53.3	0.001
Chittagong	34.3	32.3	28.2	24.9	27.4	0.002	42.9	49.5	52.9	67.6	57.7	<0.001
Dhaka	45.4	35.2	46.1	44.9	1.1	0.270	45.3	49.9	53.1	63.6	40.3	<0.001
Khulna	43.8	32.2	35.1	25.5	41.8	0.002	59.0	67.7	62.9	72.9	23.4	0.036
Rajshahi	39.6	39.4	42.8	35.3	10.9	0.506	53.7	60.6	54.5	68.6	27.8	0.002
Rangpur ^1^	-	31.8	35.5	27.7	13.1	0.317	-	59.4	65.8	77.3	30.1	<0.001
Sylhet	27.2	19.5	25.0	19.8	27.2	0.300	36.5	38.7	49.1	62.3	71.0	<0.001
Mymensingh ^2^	-	-	-	33.9	-	-	-	-	-	61.2	-	-
Place of residence												
Urban	47.3	44.9	44.3	35.1	25.9	<0.001	51.8	60.0	59.9	70.6	36.3	<0.001
Rural	37.4	29.0	35.3	31.2	16.6	0.041	45.5	50.9	53.3	65.9	45.0	<0.001
Household wealth status												
Poorest	23.1	14.7	20.9	19.7	14.8	0.842	43.4	40.3	46.8	63.2	45.9	<0.001
Poorer	29.8	22.3	30.0	28.5	4.4	0.702	36.0	50.5	49.2	65.1	81.1	<0.001
Middle	38.2	37.6	42.5	32.6	14.7	0.275	45.9	55.7	57.4	68.4	49.0	<0.001
Richer	47.4	43.6	47.2	35.3	25.6	0.005	54.1	61.0	60.0	67.2	24.2	0.001
Richest	62.7	50.5	49.8	46.3	26.2	<0.001	57.2	60.4	62.5	72.3	26.3	<0.001
Child characteristics												
Age (month)												
6–8	39.8	29.8	36.4	31.9	19.9	0.163	12.9	20.0	19.3	30.6	136.8	<0.001
9–11	43.7	32.7	34.6	32.2	26.2	0.012	31.2	48.4	43.0	55.7	78.5	<0.001
12–17	41.7	32.5	38.0	33.2	20.6	0.040	54.3	58.7	61.5	75.2	38.5	<0.001
18–23	36.0	34.7	39.6	31.5	12.7	0.291	65.6	68.6	72.9	81.6	24.3	<0.001
Gender												
Male	42.3	34.6	39.4	34.4	18.6	0.010	48.9	54.1	53.6	66.9	36.9	<0.001
Female	37.1	30.8	35.7	29.8	19.6	0.019	45.0	51.9	56.6	67.5	50.0	<0.001
Birth order												
1	43.4	37.9	43.7	35.6	18.0	0.049	53.1	60.8	59.1	69.4	30.7	<0.001
2	39.3	35.4	39.7	32.8	16.5	0.089	44.5	54.1	54.6	67.8	52.3	<0.001
≥3	36.7	26.1	27.6	27.8	24.1	0.001	43.2	45.3	50.2	64.0	48.2	<0.001
Maternal characteristics												
Age at child’s birth (years)												
<20	36.9	32.8	42.0	29.9	18.9	0.374	46.8	58.0	53.5	65.5	40.1	<0.001
20–29	40.4	32.1	38.1	32.5	19.5	0.011	48.2	53.0	55.3	68.1	41.3	<0.001
≥30	40.9	34.6	31.1	33.5	18.2	0.037	42.9	47.6	55.6	66.0	53.9	<0.001
Level of education												
0–4	27.5	18.6	25.2	23.5	14.6	0.285	38.5	43.1	44.4	59.2	53.6	<0.001
5–9	46.9	36.9	40.0	31.2	33.3	<0.001	51.9	56.2	57.3	69.5	34.1	<0.001
10+	58.3	55.9	52.2	44.5	23.7	<0.001	59.8	68.3	66.2	70.3	17.6	0.018
Occupation												
Housewife	40.1	32.2	37.6	33.4	16.7	0.030	47.3	52.4	53.7	66.4	40.4	<0.001
Working outside	38.3	39.7	37.9	30.5	20.4	0.004	45.5	60.8	59.1	68.3	50.0	<0.001
Body mass index (kg/m^2^)												
<18.5	31.3	26.7	31.4	29.1	6.9	0.863	46.1	48.9	51.6	67.9	47.2	<0.001
18.5–22.9	39.6	32.5	36.3	28.6	27.8	<0.001	45.9	55.1	54.2	65.9	43.8	<0.001
23.0–24.0	58.6	39.2	39.3	34.8	40.6	<0.001	47.3	52.3	60.2	68.7	45.4	<0.001
≥25.0	64.7	48.9	52.6	41.9	35.2	<0.001	59.1	57.2	60.2	68.2	15.3	0.011
Paternal characteristics												
Level of education												
0–4	28.1	21.0	29.0	23.7	15.8	0.439	41.1	48.1	48.0	65.4	59.0	<0.001
5–9	45.1	37.9	38.9	31.3	30.7	<0.001	47.9	54.0	56.3	65.2	36.1	<0.001
10+	65.6	52.7	52.7	47.4	27.7	<0.001	63.3	63.5	66.4	73.4	16.1	0.001
Occupation												
Service holder/businessman	50.8	40.8	44.6	40.8	19.8	0.015	55.5	57.8	60.7	68.0	22.5	<0.001
Agriculture-based worker	31.8	27.2	32.9	23.0	27.7	0.081	44.1	49.1	47.8	64.5	46.3	<0.001
Non-agriculture-based worker	42.7	33.7	38.9	32.4	24.0	0.006	51.2	52.5	56.9	67.2	31.2	<0.001
Others ^5^	32.6	23.1	29.0	27.7	15.0	0.268	35.2	52.7	51.0	68.9	95.7	<0.001

^1^ The administrative division Rangpur was created in 2010; ^2^ The administrative division Mymensingh was created in 2015; ^3^ “+” represents increase, “-” represents decrease between 2007 and 2018; ^4^
*p*-value for the Cochran–Armitage test of % change between 2007 and 2018; ^5^ Unemployed/student, retired, beggar, etc.

**Table 5 ijerph-19-00550-t005:** Trends in consumption of fruits and vegetables among 6–23 months aged children in Bangladesh, 2007–2018.

Characteristics	Vitamin-A Rich Fruits or Vegetables (%)						Other Fruits or Vegetables (%)					
	2007	2011	2014	2017–2018	% Change ^3^	*p* ^4^	2007	2011	2014	2017–2018	% Change ^3^	*p* ^4^
Overall	53.8	36.6	40.4	37.7	29.9	<0.001	46.4	17.6	19.7	26.6	42.7	<0.001
Household characteristics												
Administrative division												
Barisal	42.5	40.6	45.0	35.7	16.2	0.465	39.6	10.6	15.5	23.5	40.6	0.026
Chittagong	45.1	35.7	36.1	40.4	10.5	0.197	42.2	11.5	17.8	30.9	26.8	0.005
Dhaka	55.1	36.2	39.1	38.5	30.1	<0.001	43.1	17.9	19.1	24.3	43.5	<0.001
Khulna	56.8	35.8	53.4	24.5	56.9	<0.001	47.7	26.4	32.8	27.2	43.0	0.002
Rajshahi	66.5	39.5	38.9	37.3	43.9	<0.001	61.2	24.3	15.0	23.3	61.9	<0.001
Rangpur ^1^	-	38.9	39.6	47.0	−21.1	0.067	-	19.1	20.8	31.7	−65.6	0.001
Sylhet	42.6	31.1	44.9	36.8	13.6	0.875	33.7	17.3	22.5	22.4	33.7	0.067
Mymensingh ^2^	-	-	-	33.4	-	-	-	-	-	27.6	-	-
Place of residence												
Urban	50.9	40.2	38.2	39.6	22.3	0.001	50.7	20.7	23.6	26.3	48.2	<0.001
Rural	54.6	35.6	41.2	37.0	32.2	<0.001	45.1	16.7	18.3	26.7	40.8	<0.001
Household wealth status												
Poorest	50.9	29.2	37.9	35.5	30.2	0.002	45.2	14.5	20.0	20.4	54.8	<0.001
Poorer	50.3	33.2	41.4	35.8	28.9	0.001	38.6	17.4	17.0	24.7	36.1	<0.001
Middle	57.2	41.2	42.6	37.0	35.4	<0.001	50.3	14.3	17.0	30.0	40.4	<0.001
Richer	57.0	39.3	39.6	33.2	41.8	<0.001	47.4	20.8	21.7	26.4	44.3	<0.001
Richest	54.4	42.4	41.2	47.7	12.4	0.090	51.8	21.7	22.6	32.2	37.8	<0.001
Child characteristics												
Age (month)												
6–8	27.0	13.9	18.6	14.6	46.0	0.001	21.8	5.0	5.6	12.2	44.1	<0.001
9–11	48.3	31.8	30.6	31.0	35.9	<0.001	39.8	15.2	20.3	19.5	51.0	<0.001
12–17	59.0	40.6	46.1	43.2	26.8	<0.001	52.1	18.5	17.7	31.4	39.7	<0.001
18–23	65.9	48.3	51.0	46.2	30.0	<0.001	57.6	25.5	28.7	31.9	44.6	<0.001
Gender												
Male	53.0	36.9	38.7	40.1	24.3	<0.001	49.1	16.5	19.1	26.9	45.3	<0.001
Female	54.5	36.4	42.4	35.0	35.8	<0.001	43.8	18.8	20.4	26.3	39.9	<0.001
Birth order												
1	57.3	40.7	41.0	40.5	29.4	<0.001	48.7	19.2	20.0	30.6	37.2	<0.001
2	51.2	35.3	39.7	35.0	31.6	<0.001	46.1	17.6	18.0	24.9	46.0	<0.001
≥3	52.5	34.1	40.6	37.6	28.4	<0.001	44.7	16.3	21.3	24.0	46.3	<0.001
Maternal characteristics												
Age at child’s birth (years)												
<20	54.9	40.7	40.3	36.1	34.2	<0.001	44.0	17.9	16.6	27.3	38.0	<0.001
20–29	54.4	34.7	40.7	37.6	30.9	<0.001	48.8	16.7	19.6	25.7	47.2	<0.001
≥30	50.4	38.7	39.8	39.3	22.0	0.012	41.9	20.4	23.9	28.4	32.2	0.004
Level of education												
0–4	49.9	30.9	36.1	34.7	30.4	<0.001	39.8	15.3	16.0	21.7	45.5	<0.001
5–9	56.7	36.8	40.2	34.7	38.7	<0.001	51.1	17.3	20.2	25.6	49.9	<0.001
10+	58.3	52.4	48.8	48.3	17.0	0.014	54.3	25.5	24.7	34.6	36.3	<0.001
Occupation												
Housewife	53.2	36.2	41.2	36.9	30.7	<0.001	43.4	17.4	19.2	25.9	40.3	<0.001
Working outside	55.7	42.0	37.9	39.0	30.0	<0.001	56.7	20.0	21.5	27.7	51.2	<0.001
Body mass index (kg/m^2^)												
<18.5	54.9	35.2	36.8	34.8	36.6	<0.001	47.8	15.0	14.3	20.9	56.3	<0.001
18.5–22.9	51.7	35.7	40.5	38.0	26.6	<0.001	44.0	18.7	20.2	25.7	41.6	<0.001
23.0–24.0	56.9	42.3	45.8	38.1	33.1	0.003	43.2	18.8	26.8	29.3	32.2	0.334
≥25.0	61.0	41.0	43.0	39.5	35.2	0.002	61.9	19.9	23.0	32.2	48.0	0.003
Paternal characteristics												
Level of education												
0–4	51.8	32.6	36.0	35.3	31.7	<0.001	43.8	17.1	18.4	23.5	46.4	<0.001
5–9	52.2	35.3	42.7	35.1	32.7	<0.001	46.6	15.1	18.7	25.9	44.5	<0.001
10+	63.4	49.5	44.9	46.0	27.4	<0.001	54.3	23.8	24.5	32.9	39.5	<0.001
Occupation												
Service holder/businessman	58.8	41.0	46.1	42.8	27.2	<0.001	46.6	17.6	21.6	28.5	38.9	<0.001
Agriculture-based worker	53.1	37.2	33.9	34.7	34.7	<0.001	48.5	18.0	16.5	25.0	48.4	<0.001
Non-agriculture-based worker	52.3	33.8	40.8	37.1	29.1	<0.001	48.9	17.6	21.1	26.4	46.0	<0.001
Others ^5^	50.8	34.0	39.0	33.4	34.3	<0.001	40.7	17.0	17.9	24.1	40.9	<0.001

^1^ The administrative division Rangpur was created in 2010; ^2^ The administrative division Mymensingh was created in 2015; ^3^ “+” represents increase, “-“represents decrease between 2007 and 2018; ^4^
*p*-value for the Cochran–Armitage test of % change between 2007 and 2018; ^5^ Unemployed/student, retired, beggar, etc.

## Data Availability

The data are available upon reasonable request from the corresponding author.

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
