# Peer review of "Trends in Complementary Feeding Indicators and Intake from Specific Food Groups among Children Aged 6–23 Months in Bangladesh"

_ijerph, 2022, doi:10.3390/ijerph19010550_

Round 1

Reviewer 1 Report

The manuscript brings positive news about significant improvements in feeding indicators in Bangladesh. However, the data management and interpretation can be improved.

Statistical analysis presented in the manuscript seems quite simple, comparing results from years 2017-2018 with results from year 2007 using Cochran–Armitage test. However, the authors report difference in trends between groups (based on residence, occupation, age etc), without reporting statistical analysis used to compare CF practices across different stata.

Too much information is provided in the result section (tables) and more focus is recommended. Why not focusing to a greater extent only on the positive changes that have occurred since 2014? Other results have previously been published in an elegant way in Maternal & Child Nutrition 2018 and can be referred to in the introduction. By this there will be much more room for detailed analysis and comparison between population groups. You might identify if predictors of appropriate CF practices, these might have changed since the last analysis or identify indicators for the most successful changes.

Some minor comments:

Please write out IYCF (infant and young child feeding) when it first appears in the text (introduction, line 84)

Please check figure 1, the number of children included at each year (only provided for 2011 in the current version).

Reviewer 2 Report

see attached file.

Round 2

Reviewer 1 Report

The authors have made few changes in the manuscript, that has improved.

I have one last comment that might increase the clarity of the manuscript.

Footnotes 3 and 5 in the tables are a bit confusing ( % reduce and increase from the year 2007 to 2018, respectively). In the corresponding heading at the top of the tables where one of the columns is reduce and the other increase and then you present both - and + values in the table. . You might consider presenting change and make it clear that a - value represent a reduction and + an increase.

Author Response

Thanks very much for noting this. We have revised the tables as you have suggested.

Reviewer 2 Report

No further comments.

Author Response

Thanks very much for accepting the revision.